# Pushing the Scalability of RDF Engines on IoT Edge Devices [note 1]

**DOI:** 10.3390/s20102788

**Published:** 2020-05-14

**Authors:** Anh Le-Tuan, Conor Hayes , Manfred Hauswirth, Danh Le-Phuoc

**Affiliations:** 1Open Distributed Systems, Technical University of Berlin, 10587 Berlin, Germany; manfred.hauswirth@fokus.fraunhofer.de (M.H.); danh.lephuoc@tu-berlin.de (D.L.-P.); 2Insight Centre for Data Analytics, National University of Ireland Galway, H91 TK33 Galway, Ireland; conor.hayes@insight-centre.org; 3Fraunhofer Institute for Open Communication Systems, 10589 Berlin, Germany

**Keywords:** Internet of Things, edge device, the semantic web, RDF engine

## Abstract

Semantic interoperability for the Internet of Things (IoT) is enabled by standards and technologies from the Semantic Web. As recent research suggests a move towards decentralised IoT architectures, we have investigated the scalability and robustness of RDF (Resource Description Framework)engines that can be embedded throughout the architecture, in particular at edge nodes. RDF processing at the edge facilitates the deployment of semantic integration gateways closer to low-level devices. Our focus is on how to enable scalable and robust RDF engines that can operate on lightweight devices. In this paper, we have first carried out an empirical study of the scalability and behaviour of solutions for RDF data management on standard computing hardware that have been ported to run on lightweight devices at the network edge. The findings of our study shows that these RDF store solutions have several shortcomings on commodity ARM (Advanced RISC Machine) boards that are representative of IoT edge node hardware. Consequently, this has inspired us to introduce a lightweight RDF engine, which comprises an RDF storage and a SPARQL processor for lightweight edge devices, called RDF4Led. RDF4Led follows the RISC-style (Reduce Instruction Set Computer) design philosophy. The design constitutes a flash-aware storage structure, an indexing scheme, an alternative buffer management technique and a low-memory-footprint join algorithm that demonstrates improved scalability and robustness over competing solutions. With a significantly smaller memory footprint, we show that RDF4Led can handle 2 to 5 times more data than popular RDF engines such as Jena TDB (Tuple Database) and RDF4J, while consuming the same amount of memory. In particular, RDF4Led requires 10%–30% memory of its competitors to operate on datasets of up to 50 million triples. On memory-constrained ARM boards, it can perform faster updates and can scale better than Jena TDB and Virtuoso. Furthermore, we demonstrate considerably faster query operations than Jena TDB and RDF4J.

## 1. Introduction

The Internet of Things (IoT) proposes to connect a vast amount of everyday devices (“things”) to the Internet to enable innovative and smarter domestic and commercial services [1]. These devices range from physical objects [2] such as smart phones, smart watches, environmental sensors to virtual objects [3], such as tickets and agendas. In fact, any online service that has a unique identifier and is accessible on the Internet may be connected as a “thing” to the network. The heterogeneity of devices, services and requirements has driven major research initiatives to accelerate the real-life deployment of IoT technologies [4]. In 2019, Gartner [5] reported that 4.81 billion IoT devices were active, and predicted that the number would reach 5.1 billion by the end of 2020. In 2018, nearly two and a half exabytes of data were generated every day [6]. However, it was also observed that 80% of companies lacked skills and technology to make sense of the data provided by the IoT devices [7]. Consequently, a clear challenge is how to make the best use of the huge amount of information available from IoT networks.

It has long been recognised that standards for the integration and analysis of data must play a key role in the value generated by IoT [8]. The Semantic Web, known as an extension of the World Wide Web, aims to allow data to be interoperable over the Web [9]. Semantic technologies have been proposed to deal with data heterogeneity and to enable service interoperability in the IoT domain [10], and to underpin resource discovery, reasoning and knowledge extraction [11]. Recent research has seen the development and deployment of ontologies to describe sensors, actuators and sensor readings [12], semantic engines [13] and semantic reasoning agents [14]. These efforts have constituted important milestones towards the integration of heterogeneous IoT platforms and applications

The Resource Description Framework (RDF) (see Section 2) is now the preferred data model for semantic IoT data [15,16] and RDF engines have been used as semantic integration gateways on the IoT [17]. While existing centralised or cloud solutions offer flexible and scalable options to deal with different degrees of data integration [13], in the context of IoT, a cloud infrastructure as a single processing node may lead to network latency issues [18]. In fact, directly pushing IoT data to the cloud may have several disadvantages because it is estimated that only 10% of preprocessed IoT data is worth saving for analysis [7], as most of the intermediate data can be discarded immediately. As broadband networks have more downstream bandwidth than upstream bandwidth, large uploads of raw sensor data, for example, may quickly dominate upstream network traffic.

Furthermore, real-time IoT applications may suffer due to the resulting network latency between cloud and end-user devices. As such, it has been proposed that a decentralised integration paradigm fits better with the distributed nature of autonomous deployments of smart devices [19]. The idea is that moving the RDF engine closer to the edge of the network, and to the sensor nodes and IoT gateways, will reduce network overhead/bottlenecks and enable flexible and continuous integration of new IoT devices/data sources [20].

Thanks to recent developments in the design of embedded hardware, e.g., ARM (Advanced RISC Machine) boards [21], lightweight computers have become cheaper, smaller and more powerful. For example, a Raspberry Pi Zero [22] or a C.H.I.P computer [23] costs less than 15 Euros and is comparable in size to a credit card. Such devices are powerful enough to run a fully-functional Linux distribution that are efficient in power consumption. Their small size makes them easier to deploy or embed in other IoT devices (e.g., sensors and actuators), which provides reasonable computing resources. Furthermore, they can be placed on the network edge, as *edge devices*, i.e., data-processing gateways that interface with outer networks. For example, such an integration gateway device may be used for an outdoor ad-hoc sensor network. This gateway can be easily fitted into a lamp pole on a street or at a traffic junction, sharing a power source powered by a small solar panel.

Despite their advantages in power consumption, size and cost-effectiveness, lightweight edge devices are significantly under-equipped in terms of the memory and CPU for supporting regular RDF engines. Of the 100 billion ARM chips shipped so far [24] and even much more in the coming future, only a small fraction, e.g., 0.1%, will need a special class of RDF engine optimised for this environment. Nevertheless, this still equates to 100 million devices, which motivated us to design and build an RDF engine optimised for the hardware constraints of lightweight edge devices.

Lightweight edge devices are different from standard computing hardware in two major ways: (i) They have significantly smaller main memory and (ii) they are equipped with lightweight flash-based storage as secondary memory. To manage large static RDF datasets, the standard stand-alone or cloud-based RDF engines apply sophisticated indexing mechanisms that consume a large amount of main memory and are expensive to update. Our empirical study (see Section 3) shows that applying the same approach on resource-constrained devices causes high amounts of page faults or out-of memory errors, which heavily penalises system performance. Flash memory devices are smaller in size, lighter, more shock resistant and consume much less power than computers with disks or HDD. However, the I/O behaviour of flash-based storage, especially the erase-before-write limitation [25], degrade the efficiency of disk-based data indexing structures and caching mechanisms [26]. The majority of existing RDF engines do not cater for storage of this kind.

Inspired by the RISC design (Reduced Instruction Set Computer) of ARM computing boards, in this paper we introduce a RISC-style approach similar to [27] to address these shortcomings and others found by our empirical study. In contrast to [27], we focus on minimising memory consumption, scalability (maximising data processing) and processing performance. Our approach is based on a redesign of the storage and indexing schemes in order to optimise flash I/O. This has led to an improved *join* algorithm with significantly lower memory consumption. As a result, our RDF engine, RDF4Led, has a small code-footprint (4MB) and outperforms RDF engines such as Jena TDB and RDF4J. The experiments in Section 8 show that RDF4Led requires less than 30% of the memory used by competing RDF engines when operating on the same scale of data.

In summary, our contributions are as follows:We intensively study the scalability of the PC-based RDF engines running on IoT lightweight edge devices.We introduce a RISC-Style RDF engine design based on observations drawn from an empirical study of the performance of PC-based RDF engines running on lightweight edge devices.We develop a flash-friendly indexing data structure, a flash-friendly buffer management technique and a low-memory-footprint *join* algorithm to store and query RDF data on lightweight IoT edge devices.We implement our prototype in Java, and evaluate it to show the performance gains of our approach.

This paper is structured as follows: In Section 2, we present the fundamentals of the RDF data model, provide a short introduction to SPARQL queries, and describe how to build and RDF storage and how to query RDF data. Section 3 investigates the performance of RDF engines on standard hardware when they are ported to run on the lightweight edge devices without optimisations. Then we introduce our RISC-style approach to build an RDF engine for lightweight edge devices, called RDF4Led, and overview its architecture in Section 4. In the following sections, we present in detail the design of our flash-aware storage and our new indexing structure (Section 5), buffer management (Section 6) and the algorithm for dynamically computing joins (Section 7). In Section 8, we discuss the results of our evaluation of RDF4Led against other engines on different types of devices. Finally, in Section 9, we present our conclusions and an outlook for future work.

## 2. Background

### 2.1. RDF and SPARQL

The Resource Description Framework (RDF) is a graph-based data model that has been developed for the representation of properties and relationships of web resources. Initially, the development of RDF intended to contribute to the Semantic Web [9], however, its usage is now much wider than that. RDF is a promising standard to represent IoT data which usually refers to attributes of the phenomena observed by things and the relations among the things.

RDF presents data by using statements in a similar way to using natural language to express facts. A statement is given as a triple consisting of a subject, a predicate, and an object. The subject denotes an entity, the predicate denotes a property (relation), and the object denotes an entity or a value. Put simply, a triple states that a subject has a relation to the object or a subject has an attribute whose value is the object. Box 1 presents an RDF example describing a wind sensor W01 and its observations, i.e., *“:windSensorW01 is a sensor which observes wind speed”* and *“:windSensorW01 measured the wind speed to be 30km/h at timestamp 2019-10-03T08:04:50”*. In the example, the RDF triples are represented in Turtle—a standard format for serialising RDF data. RDF can also be serialised in other standard formats like RDF/XML or JSON-LD.

Listing 1An example of RDF statements (in Turtle format) describing a wind speed sensor and a wind speed observation.
1. 
 :windSensorW01 a sosa:Sensor;
2. 
          sosa:observes :windSpeedRate;
3. 
          sosa:madeObservation :winspeedObs01.
4. 
 :winspeedObs01 sosa:observedProperty :windSpeedRate;
5.           sosa:hasSimpleResult "30 km/h";6.           sosa:resultTime "2019-10-03T08:04:50"^^xsd:dateTime.


According to the RDF standard, in an RDF triple, the subject is an International Resource Identifier (IRI) to denote a named entity, or a blank node to express an anonymous entity. The predicate is always an IRI and the object can be an IRI, a blank node or a literal. As a consequence, there are two types of RDF triples: literal triples for describing an entity’s property and RDF links for denoting a relationship of two entities. In Box 1, the triples in line 5 and line 6 are literal triples which contain RDF literals as their objects. RDF literals can be basic or complex, e.g., integer, float or datetime, to define the values such as string, number or time. RDF links, on the other hand, consist of three IRIs. The predicate IRI connecting two entities has a type to describe the relationships between two things. These types are defined in ontologies. For example, triples in line 1-4 are RDF links, and the relationships are defined in the SSN ontology [28]. Hence, an RDF dataset or an RDF Graph is a set of RDF triples <s, p, o> ∈ (I∪B) × I × (I∪B∪L), where I, B and L are sets of IRIs, blank nodes and literals.

Listing 2An example of a SPARQL query that returns the strongest wind speed in each day of month.
1. 
 PREFIX rdf:<http://www.w3.org/1999/02/22-rdf-syntax-ns#>
2. 
 PREFIX sosa:<http://www.w3.org/ns/sosa/>
3. 
 
4. 
 SELECT ?sensor ?month ?day (max(?windSpeed) as ?maxWindSpeed)
5. 
 
6. 
 WHERE
7. 
 {
8. 
  ?sensor  a          sosa:Sensor;
9. 
       sosa:observes     :winspeedObs01;
10. 
       sosa:madeObservation ?winObs.
11. 
  ?winObs sosa:hasSimpleResult  ?windSpeed
12. 
       sosa:resultTime    ?time.
13. 
 }
14. 
 GROUP BY ?sensor (month(?time) as ?month) (day(?time) as ?day)



SPARQL is a query language which was developed to query RDF datasets. Since RDF is a graph-based data model, SPARQL is designed as a graph-matching query language. SPARQL queries include graph query patterns, conjunctions, disjunctions and optional patterns as formalised in [29]. This ability is essential for semantic integration of heterogeneous data. SPARQL also supports aggregations, filters, limits and federation, etc. Box 2 shows an example SPARQL query Q to search for the highest wind speed per day in a month measured by wind speed sensors.

A SPARQL query is of the form H ← B, where B, the body of the query, is an RDF graph query pattern matched against an RDF graph, and H, the head of the query, defines how to construct the answer of the query [29]. In Box 2, the body of Q is the text from line 8 to line 12. The graph query pattern of query Q is the matching condition to search for the RDF subgraphs containing the information of wind speed sensors and their observations. A matched subgraph is similar to an RDF graph presented in previous example (Box 1). The head of Q in line 4 indicates that the max aggregation is applied on the found wind speed values grouped by sensors, days and months.

A graph query operation is performed by matching the variables in its query pattern. In practice, this operation is expressed as a query pattern that is composed of triple patterns and joining matched triples. A triple pattern is also a variation of an RDF triple in which s,p,o can be variables. A set of triple patterns is called a basic graph pattern (BGP) and can be modelled as a directed graph. A BGP can have different shapes (e.g., a star shape, linear shape, snowflake shape and complex shape) which influences the performance of graph matching operations [30].

### 2.2. Storing and Querying RDF Data

To deal with the data heterogeneity, IoT data can be annotated with RDF by semantic gateways [31] or semantic information brokers [17]. The process of mapping IoT data into RDF, known as “semantisation”, consists of three steps: collecting sensor data, enriching the raw data and generating RDF statements to semantically annotate the data [15]. Therefore, it is essential to manage the RDF data generated from the IoT mediator nodes efficiently.

The goal of RDF data management is to facilitate efficient storage and query processing of RDF data. RDF data management has received considerable attention within the Semantic Web community. As a result, there are many works focusing on RDF storage and SPARQL query processing [32]. An RDF engine, which contains an RDF storage and a SPARQL processor, can be classified into two types: non-native RDF engine and native RDF engine.

A non-native RDF engine is built on top of a traditional database management system [32]. For example, 3store [33] and Oracle [34] use components of a relational database management system to build their RDF storage and SPARQL query processor. In these systems, RDF statements are stored in a single table of three columns corresponding to three constituents of an RDF statement (s,p,o). An index is added to each column to speed up lookup operations. This approach, known as the triple-table approach, scales poorly due to many self-joins over this single, possibly very large table when executing complex queries [35]. Therefore, the so-called “property table technique” was introduced to reduce the number of self-joins. Here, RDF statements are stored in many tables and each table includes a subject and several predicates. Therefore, triples that have the same subject can be retrieved without an expensive join operation. Sesame [36], Jena2 [37] are among the RDF engines using this approach. An alternative to the property table approach was suggested in SW-Store [38] which organises RDF datasets in two-column tables. Each table contains the subject and object of the triples which have the same predicate. The tables are stored in the column-oriented database C-store [39]. Moreover, there have been several works to store RDF data in distributed database systems and in cloud infrastructures. For example, Jena-HBase [40] integrates Jena and Apache Hadoop [41], whereas AMADA [42] is an RDF data management platform based on Amazon Web Services.

A native RDF engine, on the other hand, is designed and optimised from scratch to manage RDF data. Instead of adapting RDF concepts into the concepts that are native to the underlying database systems, a native RDF engine is fully optimised for persisting and querying RDF data. Naturally, the design of native RDF storage is heavily influenced by traditional database design. For example, Virtuoso [43] or 4Store [44] store RDF statements in a table-like structure. In these systems, RDF data is presented as “RDF quads” consisting of 4 elements: subject, predicate, object and graph id (or model in 4Store). Each of these attributes can be indexed in different ways to improve query execution performance. Other RDF systems, e.g., YARS(Yet Another RDF Store) [45], Hexastore [46] and RDF-3X [27], employ index-permuted storage systems. In particular, the string representations of RDF resources are replaced with unique integers. Indexes are built on the shorter encoded values and cover all major accesses types to triple query patterns. As the RDF data model is a graph data model, it was also suggested to store and process RDF data by graph data structures and algorithms. The systems following the graph-based idea include TripleT [47], DOGMA [48], Diplodocus [49]. However, many graph algorithms are known to be complex in terms of implementation and optimisation.

Most of these solutions for RDF data management focus on scalability in dataset size and query complexity for standard computer hardware and cloud infrastructures equipped with large main memory and multiple disks. Their efficiency and scalability is achieved by choosing appropriate index structures and join techniques for large main memory machines [50], whereas, IoT edge devices are characterised by small memory and flash-based secondary storage. Our previous results [51,52] have already indicated that, on the resource-constrained computers, these engines suffer from performance issues, pointing to the requirement to build a customised RDF data management for this type of devices. There have been several efforts to move RDF data processing to small devices. For instance, Mobile RDF [53], a lightweight RDF framework, provides simple APIs for creating, parsing and serialising RDF data. However, RDF graph modifications are not supported (no update functionality). AndroJena [54] is an adoption of Jena that offers all functionalities of the original Jena framework. However, the implementation of AndroJena has ignored the fact that RDF data processing on resource-constrained devices has different requirements which results in significant scalability issues. The second version of RDF On-The-Go [55], which was our first effort to build a native RDF storage for Android OS, can store up to 5 million triples on a common hardware configuration Android phone. Two recent notable works are the Wiselib tuplestore [56] and μRDF Store [57] designed for very constrained memory devices which are intended to store only a few thousands of RDF triples.

## 3. Empirical Study

In the current state of the art, many RDF engines can manage billion to trillion triple datasets [32]. To achieve this scale, these RDF engines must be executed on powerful computers equipped with hundreds of GBs of RAM and many CPUs. Clearly, these approaches suffer from performance drawbacks on resource-constrained IoT edge devices. In addition, these RDF engines are optimised for storage and querying of large, heterogeneous, and rather static RDF datasets, whereas IoT data is more dynamic and less heterogeneous. In this section, we study the performance drawbacks of RDF engines that were directly ported to run on lightweight edge devices. The findings of this study are the input for the design of our RDF engine optimised for lightweight IoT edge devices presented in Section 4.

The empirical study is conducted in a simulated scenario of a weather data management system specifically created for the IoT domain. To enable semantic interoperability in this IoT system, the weather data is described in RDF and can be queried with SPARQL. In the IoT domain, RDF is often used to semantically annotate the metadata of IoT platforms, systems and devices such as the location of an IoT device or the specifications of sensors deployed on an IoT platforms; the metadata of observations such as types of the observations or units of the measurement; and observation timestamps and actual readings.

Traditionally, an IoT system that manages semantic weather data can be deployed in a centralised fashion with a 3-layer architecture. At the lowest layer, the IoT devices are the wireless sensors or actuators deployed to collect environment data such as temperature, humidity, etc. The collected data can be transmitted wirelessly to the second layer via a number of wireless protocols. The middle layer consists of the IoT gateways functioning as the protocol translators that transmit the collected data to the upper layer. Via long-range wired or wireless networks, the data then can be transferred to the third layer which can be powerful servers or cloud infrastructures. At the third layer, the collected data can be mapped to RDF. This layer can also provide SPARQL endpoints to allow users to query the RDF data.

However, it is argued that to reduce the network traffic, to scale up the system, and to support real-time operation, the data should be preprocessed (i.e., annotated, aggregated and filtered) and queried on the IoT gateways themselves [31]. This means that RDF engines for these gateway devices must exist and be efficient on their hardware. Hence, the raw data can be mapped to RDF and can be queried by using SPARQL query from the middle layer. In this study, we set up RDF engines on resource-constrained single board computers that are representative of IoT gateway hardware configurations. We use sensor readings from the Integrated Surface Database (ISD) of the National Climate Data Center (NCDC) as sample data. The data is mapped to RDF and is stored locally on these devices. On each device, for each RDF engine, we test how much RDF can be stored and how fast the data can be inserted and queried. The details of the hardware configurations of our testing devices, RDF engines, RDF schema, testing SPARQL queries and the experiments are presented in the following sections.

### 3.1. Hardware Devices

Recent technological advances of embedded processors have increased the processing capabilities of IoT edge devices. Based on their computational capabilities, IoT devices can be categorised into low-end devices and high-end devices. The low-end IoT devices which are very constrained in terms of energy, CPU (less than 100MHz), and memory capacity (less than 100 kB) resources. Popular examples of devices in this category include Arduino [58], Zolertia [59], OpenMote node [60], etc. The second category consists of high-end IoT devices which include single-board computers such as Intel Galileo [61], Raspberry Pi [62], Beagle Bone board [63] or smartphones. They have enough resources and adequate capabilities to run software based on traditional operating systems such Linux or BSD (Berkeley Software Distribution).

We conduct our empirical study on five types of high-end IoT devices: Intel Galileo Gen II (GII), Raspberry Pi Zero W (RPi0), Raspberry Pi 2 version B (RPi2), Raspberry Pi 3 (RPi3), and Beagle Bone Black (BBB). They were chosen because of their popularity and thus proof of concept in the IoT domain. Furthermore, they are good representatives of resource constraints of IoT gateways. The configurations of each device used in the experiments are summarised in Table 1.

The Intel Galileo [61] was developed by Intel and was the first IoT device that could run a complete Linux system. The Intel Galileo was designed as to be Arduino-compatible which enables Arduino sensors and actuators to be used out of-the-box. Therefore, it can be considered to be composed of two sets of hardware components, i.e., the Arduino hardware and the “mini PC” hardware. The Intel Galileo is equipped with RAM, Flash memory, a mini SD card reader, an Ethernet adaptor and a Intel Quark x86 CPU. Galileo came out in two versions, Gen 1 Intel Galileo Gen 2 Intel Galileo, which are very similar with the Gen 2 boards having some improved hardware. In the experiments we use only Galileo Gen 2 hardware.

BeagleBoard [63] was originally developed and introduced by Texas Instruments in 2008. By using the OMAP3530 system-on-a-chip technology, the BeagleBoard board is a good platform for various demonstration scenarios in the IoT world and it was regarded as a giant step to bring the microcontrollers to a fully-fledged microcomputer. BeagleBone Black is a small credit card-sized board that was launched in 2013 at a price of 55 EUR. Despite its small size and low cost, the BeagleBone Black is equipped with 512 MB RAM, a 1 GHz clock ARM Context-A8 processor and 2 GB eMMC flash memory.

The Raspberry Pi [62] is another well-known, low-cost type of single board computer that was developed by the University of Cambridge. The Raspberry Pi Zero was introduced in 2015 and features an ARM Cortex A8 single core CPU with 1.0 GHz and 512 MB RAM. In 2017, the Raspberry Pi Zero W was launched, a newer version of the Pi Zero with Wi-Fi and Bluetooth capabilities. The Raspberry Pi 2, which is more powerful than Pi Zero, has an ARM Cortex A7 CPU with a 0.9 GHz CPU cycles/core, quad-cores, and 1 GB of RAM. The Raspberry Pi 3 is equipped with a 1.2 GHz 64-bit quad-core ARM Cortex-A53 processor, with 512 KB shared L2 cache, and 1 GB of RAM. The Raspberry Pi runs Raspbian OS, a free operating system based on Debian Linux and optimised for the Raspberry Pi hardware. Despite the availability of 64-bit CPUs on Raspberry Pi 3, at the time this work was done, Raspbian OS with 64-bit has not released yet.

### 3.2. RDF Engines

We selected three RDF engines that can be set up on the above devices: Apache Jena, RDF4J and Virtuoso. The technical specifications of each engine used in this study are reported in Table 2.

Apache Jena [64] is a well-known open-source framework for RDF data processing implemented in Java. To support persistent RDF storage, Apache Jena provides a native RDF storage called Jena TDB (Tuple Database). Jena TDB stores RDF terms (nodes) in a node table and employs multiple indexes for RDF triples. The data in the node table and the indexes are organised on fixed length key and fixed length value B${}^{+}$Trees. Jena TDB is able to operate on both 32-bit and 64-bit systems. However, it performs better on a 64-bit machine because its caching mechanism requires more memory. On a 32-bit JVM, the size of dataset might be limited because Java addressing cannot grow beyond 1.5 GB. To deal with this limitation, TDB employs an in-heap LRU cache of B${}^{+}$Tree blocks. Thus, it is recommended to configure JVM with at least 1GB for Jena TDB to achieve the sufficient performance.

RDF4J [65](formerly Sesame [36]) is another RDF data processing framework implemented in Java and is available as open-source software. Native Store is the persistent RDF storage of RDF4J. In addition, this component communicates with other stores via SAIL API (Storage and Inference Layer). Native Store is designed to support medium datasets (e.g., 100 million triples) on common hardware. It uses direct disk I/O and employs on-disk indexes to speed up queries. Again, B-Trees are used for indexing RDF statements and each RDF statement is stored in multiple indexes. By default, the Native Store uses two indexes: subject-predicate-object-context(spoc) and predicate-object-subject-context. Indexes can be added or dropped on demand to speed up querying or saving disk space.

Virtuoso [66] is developed by OpenLink Software Inc, and is available both as an open-source and a commercial version. Different from the other engines, Virtuoso uses a relational database back-end storage to store RDF, and is implemented in C++. Virtuoso is well-known as a traditional relational database supporting RDF data and as a SPARQL-to-SQL solution to manage RDF data. The older version of Virtuoso stores RDF data in a row-wise format storage. Meanwhile, column-wise format storage has been adopted since version 7. In a row-based storage, RDF datasets are stored as collections of RDF quads that consist of graph ids, subjects, predicates and objects in a single table. From version 7, Virtuoso only operates on 64-bits OS. Thus, we have compiled and set up the Virtuoso 6 open-source version in the evaluation.

### 3.3. Weather Dataset and RDF Schema

As mentioned above, we use the ISD (Integrated Surface Dataset) dataset [67] as the sample dataset for our experiments. The ISD dataset is one of the most prominent weather datasets that contains weather observations collected from 20 thousand weather stations from all over the world since 1901. The observations include the measurements such as temperature, wind speed, wind angle, etc. Additionally, the observations also contain the timestamps when these measurements were made.

To describe the metadata of the weather stations such as location, deployed sensors and the observation in RDF, we use the Semantic Sensor Network (SSN) ontology, the Quantity Kinds and Units (QUDT) ontology, as well as Geo Name (GeoNames) and Basic Geo (WGS84). Figure 1 illustrates a sample of the RDF schema used to describe the metadata of the weather stations in the ISD dataset.

In the ISD dataset, each weather station is assigned a unique station ID. Thus, to create a unique IRI to refer to a weather station, we concatenate a unique prefix and the station ID. For example, the IRI *station:001001* is used to refer to the weather station 001001. Following the specification of the SSN ontology, a weather station is described as a platform that hosts multiple sensors or devices. For instance, the IRI *station:001001* is described to have type *sosa:Platform* and host a sensor whose resource is *sensor:001/temp*. The location of the station is described by using GeoNames and WGS84. The class *sosa:ObservableProperty* is used to define the phenomenon and property that a sensor can observe. For instance, the resource *:Temperature* refers to an observable property which is the temperature. The temperature sensor *sensor:001/temp* is described as a sosa:sensor that observes the *:Temperature*.

The RDF schema of a sensor reading is shown in Figure 2. A resource referring to a sensor reading is described as an observation by using the *sosa:Observation* class. The type of the observation is expressed by using the *sosa:observedProperty* property. Defining an observation and its type is similar to how a sensor and its observable phenomenon. For example, the temperature observation *Observation/001* is defined as a *sosa:Observation* whose observed property is *Temperature*. The timestamp of an observation is defined by using the Date and Time data type of the XML Schema Definition Language(XSD) and assigned to the observation with the predicate *sosa:resultTime*. The actual reading from an observation is described with the predicate *sosa:hasSimpleResult* or more explicitly with the predicate *sosa:hasResult*. In this example, the unit of the temperature reading is presented in Celsius degree. Furthermore, the *sosa:FeatureOfInterest* vocabulary is used to enhance the expressiveness of an observation. For instance, the observation in the example is known as a temperature observation at location *location/001*. With this RDF schema, approximately 80–90 RDF triples are required to map an observation from the ISD dataset to RDF.

The queries used in the experiments were created following the design of the WATDIV benchmark [30] to test the performance of each SPARQL query processor against different shapes of BGP. Eleven query templates were created and can be found in our Github repository (https://github.com/anhlt18vn/sensor2020). The query templates are categorised into three groups: linear(L), star (S) and snowflake (F). The BGPs of the query templates in the linear group contain multiple low degree join triple patterns, whereas, the BGPs in the star-shaped queries are composed of single high degree join triple patterns. The snowflake-shaped queries have a mix of low degree joins and high degree joins.

### 3.4. Experiment Design

We evaluated the selected RDF engines with three experiments. First, we test the update throughput of each engine on each device. As presented in our scenario, on the high-end devices, these RDF engines may serve for embedded semantic data management that supports the semantic gateway services [31] or semantic information brokers [17] architectural styles. Hence, they are required to deal with dynamic data flows from the sensors. Then, we test the query response time. Finally, we measure the memory consumption of the RDF engines when these engines perform storing and querying operations.

Figure 3 depicts the setup of our experiments. The ISD data is read and mapped into RDF with the RDF schema described in Section 3.3 by the *ISD-2-RDF Wrapper*. The processes of inserting and querying the generated RDF data to/from the *RDF Engine* are managed by the *RDF Data Insert and Query Monitor*. This component is also responsible for recording the performance of these processes. The complete source code of the implementation of our experiments can be found in our Github repository (https://github.com/anhlt18vn/sensor2020).

Exp1—Update throughput: The first experiment is to test how much new data the system can incrementally update with a certain underlying RDF store corresponding to each hardware configuration. We simulate a process of data growth by gradually adding more data to the system. We measure the rate of inserting data (triples/s) and query response time the until the system crashes or until the speed is below 80 triples/s (whichever happened first). If the system cannot update 80 triples/s that means it is not able to update one observation/s. We extract data from 25 weather stations from the ISD dataset in the last six months of the year 2019. The number of the observations is approximately 600 thousands, and the size of the generated RDF dataset is about 50 million RDF triples.

Exp2—Query evaluation: In the second experiment, we test the query response times of each engine. On each device, we choose the dataset with a scale which all the engines can store. For each dataset, from each query template, we generate 100 queries. We record the maximum, the minimum and the average time that these engines need to answer each type of query. The generated queries for each dataset are 1100 (11 query templates) in total.

Exp3—Memory consumption: In the third experiment, we measure the memory consumption of three system configurations, when they perform the insertion and query. The experiment runs the queries repeatedly and records the maximum memory heap that the operating system allocates. Note that the memory consumption is device-independent. To evaluate the impact of the data size on memory consumption, the test is conducted on the Raspberry Pi 3 with ten different sized datasets and 10 set of queries according to each dataset. The scale ranges from 5 million triples to 50 million triples.

### 3.5. Experiment Report and Findings

Figure 4 illustrate the results of **Exp1**, in which we measured the update throughput of Virtuoso, Jena TDB and RDF4J on five types of the lightweight computing devices. On GII, RPi0 and BBB, none of the RDF engines could finish inserting the dataset of 50 million RDF triples. They crashed in the middle of the test due to an “out of memory” error. For instance, on GII (see Figure 4a), Virtuoso was able to insert 9 million RDF triples, Jena TDB and RDF4J stopped after inserting 5 million RDF triples. Due to the similar hardware settings, the scalability behaviours of these RDF engines on RPi0 (see Figure 4b) and on BBB (see Figure 4c) were similar. On both devices, Virtuoso could store 40 million RDF triples, whereas Jena TDB and RDF4J were only able to store 20 million RDF triples. On the other hand, on the RPi2 and RPi3, with more powerful computational capabilities, all three RDF engines could finish the test and store up to 50 million RDF triples (see Figure 4).

In general, the update throughput of the three engines decreased when the size of their storage increased. On GII, after inserting 3 million triples, the update throughput of Virtuoso was 250 triples/s (approx. 3 observations per second) whereas Jena TDB and RDF4J only could insert 50 triples/s (less than 1 observation per second). Before crashing, Virtuoso’s update speed was less than 100 RDF triples per second, whereas the update speeds of Jena TDB and RDF4J were only 20 and 50 triples/second respectively. On RPi0 and BBB, the inserting speed of Virtuoso was up to 600–900 triples/s in the first 10 million triples. However, Virtuoso’s speed dropped dramatically, when storage size reached 15 million triples. Its speed remained 350 triples/s, which is only half of its peak speed. The update behaviour of Jena TDB and RDF4J was similar to that of Virtuoso. However, with the same storage size, the speed of Jena TDB and RDF4J was less than half of the speed of Virtuoso. On RPi2 (see Figure 4d) and RPi3 (see Figure 4e), the insertion throughput of these RDF engines was much higher and dropped slower than that on RPi0 and BBB. At the beginning of the test, Virtuoso inserted data with a speed of up to 1300–1400 triples/s. The insertion speed of Virtuoso decreased to 700–900 triples/s later when the storage size was up to 40 million RDF triples. Again, insertion speed of Jena TDB and RDF4J on RPi2 and RPi was 2–3 times slower than that of Virtuoso.

Figure 5 reports the results of the **Exp2** in which we compared the query response time of the RDF engines. On each type of devices, we used the datasets that all the engines could handle. For instance, we used datasets of 5 millions, 20 millions and 50 millions of RDF triples respectively to conduct the second test on GII (see Figure 5a), BBB (see Figure 5b) and RPi3 (see Figure 5c). In general, all the RDF engines could answer the tested SPARQL queries. Among the three RDF engines, Virtuoso was always the fastest to return the answer for every query. In most cases, Jena TDB and RDF4J were able to answer these queries in less than 10 seconds. However, to answer the more complicated queries, e.g., snow flake shape query F3, took roughly a minute.

The difference in scalability and performance of the three engines can be explained by their memory usage which is reported in Figure 6a,b. Note that, a part of the memory is occupied by the operating system. Therefore, the maximum available memory for the applications is always lower than the size of RAM. For instance, there is only 230 MB available memory on the GII, nearly 380 MB on P0 and BBB and 950 MB in RPi2 and RPi3. The memory consumption of Jena TDB and RDF4J gradually increased according to the size of the storage. In the throughput test (see Figure 6a), the memory consumption of Jena TDB and RDF4J rose up to 230 MB, 380 MB and 650 MB after inserting 5 million, 20 million, 50 million RDF triples, respectively. The memory usage histogram of Jena TDB and RDF4J explains why they ran out of memory when operating on GII, RPi0 and BBB. In contrast, the memory buffer of Virtuoso was statically set depending on the maximum RAM available on each device. For instance, it was set to 200 MB on GII, 350 MB on BBB and RPi0 and 850 MB on RPi2 and RPi3. On the same device, Virtuoso had better scalability than Jena TDB and RDF4J because it handled the buffer memory better. For example, writing data from memory to the secondary storage to claim back memory space could help Virtuoso enlarging its storage. Virtuoso was able to store up to 8.5 million RDF triples on GII, and near 40–42 million RDF triples on RPi0 and BBB. Writing data into the secondary memory to create more room for caching new data is a widely used technique in conventional database management systems [68]. By using this technique, it can be explained why the insertion speed of Virtuoso dropped dramatically and was heavily penalised on the devices with less memory. However, compared to Jena TDB and RDF4J, Virtuoso used bigger buffer memory to cache more data in main memory, which explains why in our experiments Virtuoso could update data and answer the queries faster.

## 4. RISC-Style Approach for Lightweight Edge Devices

### 4.1. Rationale of Our System Design

From our empirical study, we clearly see that lightweight edge devices are different from standard computing hardware in respect to two characteristics that determine the performance of an RDF store: (i) They have a significantly smaller amount of main memory and (ii) they are equipped with flash-based storage as secondary memory and storage. Besides that, data processing on the network edges operates in a dynamical environment with frequent data updates and changes in devices and sensors.

Typically, RDF engines are optimised for machines with massive amounts of RAM and multiple high-performance disk arrays. This abundance of resource enables them to store billion-triple datasets and answer complex SPARQL queries. However, our empirical study has shown that these RDF engines suffer from significant performance problems when they run on the edge devices. To manage large static RDF datasets, these RDF engines use sophisticated indexing mechanisms that consume huge amounts of main memory and incur high update costs. Using too much memory on memory-constraint devices may cause system paging behaviours or out-of-memory errors that heavily penalise the performance or harm the robustness. Such inefficiency is due to the lack of main memory, and a less efficient disk-based data indexing structure and caching mechanism on flash-based storage.

In comparison to hard disks, flash-based storage devices have faster random accesses but fail to provide fast random writes [69]. Flash memory stores information in arrays of semiconductor memory cells which are organised into pages and pages are grouped into blocks. A page is the smallest unit that can be read or written with flash memory. On flash memory, updating a single page in a block is not possible. Instead, first the the whole block must be erased and then the updated data can be written to this block. Erase is an operation particular to flash-based memory. Thus, write-in-place operations, that update a single piece of data in a block, consists of two operations on the entire block: erase and write. Common indexing techniques which are designed for magnetic disks do not manage well this erase-before-write limitation. As a result, they suffer from slow write performance when managing data on flash memory [25]. For instance, the commonly used B${}^{+}$Tree indexing structure in RDF triple stores is does not work well on flash-based storage [70]. However, the performance of random write can be improved by aligning writes to blocks [25] and applying appropriate buffer management techniques [71].

Te RISC-style design philosophy described in [27] implements the features necessary for an RDF engine around data access and join operations. On top of that, processing loads and resource consumption are mainly caused by these operations. Thus, we will focus our design efforts on efficient components for these operations and use simple implementations for the rest with the purpose of reducing software size.

### 4.2. Architectural View

In general terms, the architecture of an RDF engine can be views a shown in Figure 7. At the bottom layer, an RDF engine has a PhysicalRDF
Storage functioning as the secondary memory to store persistent data. The Physical RDF Storage is often coupled with a Buffer
Manager to manage in-memory data. The Buffer Manager caches the data in-use to reduce disk accesses when writing to the Physical Storage or being read by the Query
Executer. Typically, an RDF engine will use a Dictionary to translate the string-based RDF resources identifiers into encoded identifiers in the form of integers or longs. The Dictionary is often coupled with an Input handler, a Query
Parser to encode the RDF resources in RDF documents or SPARQL queries, and with an Output handler to return the original form of RDF resources. This technique reduces the storage space required for RDF triples and makes comparisons (for joins) more efficient.

For our RDF engine, RDF4Led, we reuse the same architecture of traditional RDF engines. We reuse the Dictionary techniques to transform RDF resources into encoded integers. The string representations of the RDF resources are kept separately on the flash memory. The key components that differentiate our approach from traditional RDF engines are the Physical RDF Storage, the Buffer Manager and the Query Executor. They are specifically optimised for lightweight edge devices.

In an RDF engine, the algorithms and techniques, that comprise the Physical Storage, the Buffer Manager and the Query Executor, have to be optimised to the nature of the data (in this case RDF) and the particular hardware of the machine it runs on [35,68]. To reduce storage space, we used a very compact format to store a list of RDF triples known as an “RDF molecule.” To adapt to the specific flash I/O behaviours, the molecules are organised into block units whose size is equal to the flash-erase block size. On top of that, we used an in-memory caching mechanism to cache the atomic data and to cluster the write operations in order to improve the write performance. To reduce the memory required to maintain the indexes of data in the flash storage, we used an alternative index structure that is based on the Block Range Index(BRIN) approach [72]. The basic idea of BRIN is to summarise the information of a data block on persistent storage (e.g., its location) into a small tuple. The result is that we can minimise the amount of memory required to maintain the indexes.

The results of our empirical study indicate that managing memory usage of an RDF engine is the key factor to achieve robustness and scalability. The Buffer Manager is used to buffer updated data or to cache data read from Physical Storage. Its primary role is to keep the engine from crashing by unexpected out-of-memory exceptions. When needed, it flushes data to the Physical RDF Storage to reclaim free memory. Write operations are prioritised by a buffer replacement policy designed to reduce the number of overwrites on the same data block as well as the number of read operations from the Physical Storage.

To answer a SPARQL query, it is required to perform graph matching operations. These operations are the joins of the RDF triples that match triple query patterns. Among the operations to answer a SPARQL query, the graph matching operation is the most resource-intensive one. To reduce computational cost, our approach is to avoid caching the intermediate results of joins. The Query Executor uses a nested execution model [73] to join and processes the join in one-tuple-at-a-time fashion[74]. In each run, the Query Executor adaptively chooses the next triple pattern to probe and scan. The Buffer Manager is also tightly coupled with the Query Executor to provide cached data in the buffer for the efficient use of memory.

## 5. Storage and Indexing

### 5.1. Storage Layout

Our RDF storage and indexing model combines the permuted index and molecule-based storage model. We use 3 permuted indexes: SPO (subject-predicate-object), POS, and OSP, which is sufficient to cover all possible query patterns. For example, the SPO layout can be used for triple query patterns with a bound subject (s ? ?) and bound subject-predicate (s p ?). Although, using all six possible permutations combinations may answer complex queries more effectively, using only three consumes less storage space and decreases the cost of updates, i.e., we must update only three data structures instead of six, which is crucial for flash storage.

Our design consists of a *Physical Layer* and a *Buffer Layer*. The Physical Layer stores data directly on flash storage (Physical RDF Storage) and the Buffer Layer operates in main memory (Buffer Manager) and has the following roles: (i) grouping and caching atomic data updates before writing a block; (ii) indexing the data stored on the Physical Layer; and (iii) caching recently used data for read performance. This allows us to group multiple updates within a block into one erase-and-write operation and to improve read performance through the cache.

**Physical Layer:** To achieve high compression of triples on the flash storage, we leverage the molecule-based storage model. RDF molecules are a hybrid data structure. It stores a compact list of properties and objects related to a subject which is the root of the molecule. Molecule clusters are used in two ways: to logically group sets of related resources, and to physically co-locate information related to a given subject. Physically we represent a molecule as a list of co-located integers corresponding to S, P, and O as shown in Figure 8. By this, we avoid storing repetitive values multiple times. Moreover, we enable further data compression, e.g., by storing deltas of sorted integers instead of full values.

In the Physical Layer, we store sorted molecules into continuous pages (read units) which are grouped into blocks (erase units). Moreover, all entities in molecules are also sorted to improve search performance. In the example shown in Figure 9, each block in the Physical Layer contains four pages and each page stores molecules.

**Buffer Layer:** Similarly to the idea of BRIN, the Buffer Layer summarises the information of the data in the Physical Layer. In the Buffer Layer, to keep the reference to a data page, we cache the first triple of the first molecule in the page. The reference to a data block is the first page of the data block. Figure 9 depicts an example of the indexing structure that we use for storing an SPO index. The Buffer Layer maps the references of the pages and the blocks to their physical addresses. Thus, it acts as an index of the data that locates in the Physical Layer. We distinguish three types of entries in the Buffer Layer: triple entry, page entry and block entry. A *page entry* is an entry that refers to the beginning of a page in the Physical Layer, it contains the first triple in the first molecule of a page. A *block entry* is a page entry with an extra field indicating that this page is the first page a block. A *triple entry* contains an atomic triple and a value indicating whether this triple has been modified. In Figure 9, the grey columns represent block entries and the white columns represent page entries. For fast lookup operations on the Buffer Layer, all triples are sorted lexicographically. Moreover, we maintain the logical order of the triples as in the Physical Layer. This allows us to group and commit sequential pages containing modified triples and belonging to the same block in one write operation.

### 5.2. Index Lookup

To efficiently retrieve RDF triples that match a triple pattern, we execute a lookup on the index layout in which triples are stored sorted on the bound elements of the triple pattern. For example, a search for the matched triple of a triple pattern (s, p, ?) is executed on the SPO index layout, or the triples that match pattern (?, p, o) are answered with POS index layout. Thus, we can answer a triple pattern query by using a single range scan on the corresponding index layout. For instance, in the sorted SPO index layout, the matched triples of the triple pattern (s, p, ?) are of a form (s, p, $o_{i}$) and are located between (s, p, $o_{min}$) and (s, p, $o_{max}$), where $o_{min}$ and $o_{max}$ are the smallest and the highest identifiers within the layout. As triples are sorted, the matched triples are retrieved as a sublist. The sublist is computed by finding the lower and the upper bound positions of the matched triples on the layout. In this example, the lower bound position of the matched sublist is the position of the triple (s, p, $o_{min}$) and the upper bound position is the position of the triple (s, p, $o_{max}$). The matched triples are extracted from the layout by probing triple by triple from the lower bound position to the upper bound position.

### 5.3. Writing Strategy

The write operations from Buffer Layer to Physical Layer are optimised is for flash memory by following the basic principles: (i) minimise the number of physical writes to physical storage; (ii) group multiple updates into one write operation; (iii) keep a relatively high hit ratio for the data in the buffer. We use the Buffer Layer to delay write operations and to group many updates in blocks in order to mitigate the issue of single erase-before-write operations. For a high cache hit ratio, we keep the data blocks with higher chances to be accessed and modified in the future. The details of the buffer eviction strategy used in RDF4Led are presented in Section 6.

Figure 10 provide an example to illustrate the process of inserting 5 triples (03, 06, 01), (03, 06, 04), (18, 10, 12), (26, 24, 04), and (30, 28, 21). Note that, these triples contain only integers as they are already translated by the Dictionary. We first add them to the Buffer Layer. The page entries in the Buffer Layer indicate the pages where the triples will be physically inserted. For instance, the arriving triples (03, 06, 01), (03, 06, 04) are lexicographically smaller than the triple (08, 07, 09) which is the first entry of page 2. Thus, they are added to page 1. Similarly, the incoming triples (18, 10, 12), (26, 24, 04) and (30, 28, 21) will be added to page 10, page 4, and page 5.

When the size of the Buffer Layer reaches its limit or when data needs to be moved out to free memory, the triples are written to the Physical Layer. All dirty pages, i.e., pages containing modified triples, within the same block are written at the same time to save expensive erase operations. For example, when the system needs to write 2 triples from buffer to storage, we move either data belonging to block 0, i.e., triples (03, 06, 01), (03, 06, 04), or data belonging to block 1, i.e., triples (26, 24, 04), (30, 28, 21), so that only one block is erased. In case we choose triples (18, 10, 12) and (30, 28, 21), the system has to erase two blocks before writing data. Moving data from the Buffer Layer to the Physical Layer is an essential optimisation problem of maximising the number of triples we can write within a minimal number of blocks in order to minimise the number of erase operations. Obviously, block 0 is written before block 1 as its density is greater than that of block 1. The higher the density a block has, the less chance the next arriving triple will have to be inserted into that block.

## 6. Buffer Manager

Similar to a conventional database system, the Buffer Manager is responsible for handling all requests for data blocks. If the requested data block already exists in main memory, it passes the block’s reference to the requester. If not, it fetches the data block from the flash drive (Physical Layer) and loads it into main memory (Buffer Layer). It also decides which data block will be kept in main memory. To perform efficiently, the Buffer Manager requires a suitable replacement strategy to create space for a new data block and write back to Physical Storage data blocks which are no longer needed.

Our Buffer Manager uses AD-LRU [71], a flash-friendly data buffering technique. We organise the data blocks in two queues, the hot queue and cold queue (see Figure 11). In the hot queue, we keep the blocks that have been recently accessed. When releasing memory, we will not write them to the Physical Layer immediately, but to the cold queue. We organise data in a flat fashion in the hot queue. We keep sorted triples instead of molecules to speed up atomic updates as well as index look-up operations. To save memory, the cold queue contains the molecule blocks (blocks in compressed form) that are ready to be written to the Physical Layer.

Algorithm 1 illustrates the process of accessing a data block from the Buffer Manager. For requesting a given data block, the requester sends its block entry (of the Buffer Layer) to the Buffer Manager. First, the Buffer Manager looks up for the reference to the requested data block in main memory (line 2). If the requested data block does not exits in main memory, the Buffer Manager checks if the there is sufficient memory available for holding a new data block (lines 4–6). During the process of checking memory availability, exiting data blocks in the memory may be written to the Physical Layer to reclaim space for new blocks (see Algorithm 2). After having been read, the new data block is “decompressed” into its flat form, i.e., decomposed into individual triples, and added to the hot queue, while its reference is returned (lines 8–12). If the data block already existed in main memory, the Buffer Manager will check if it is in the the hot queue or in the cold queue. Note that, the organisation of the hot queue and the cold queue is similar, the only difference is that hot queue holds uncompressed data while cold queue holds compressed data. If the data block is in cold queue, the system again prepares space for decompressing the data block (lines 15–18). Before returning the reference of the data block, its position in the hot queue is adjusted to mark the data block as being recently accessed (lines 20–21).
**Algorithm 1:** Access to a data block from buffer
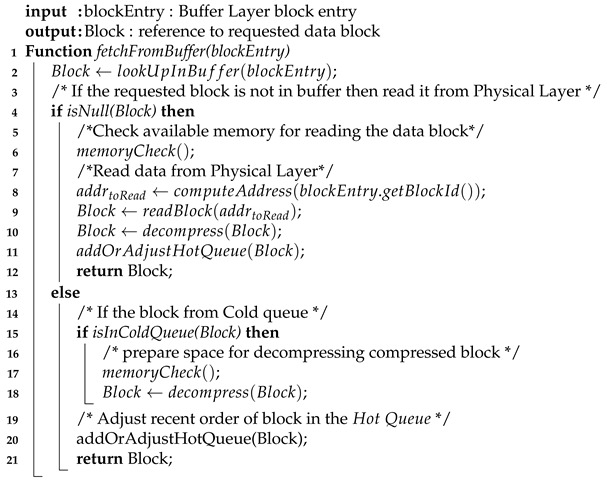


Algorithm 2 illustrates the memory checking procedure which is called in the first algorithm (line 6 and line 17). This algorithm guarantees that the system does not run out of memory during runtime. As we divide the buffer into the hot queue and the cold queue, the hot/cold ratio defines how many blocks are held in each queue. Depending on the current hot/cold ratio, the Buffer Manager decides to move a data block from the hot queue to the cold queue, or to evict a data block from the cold queue and write it to the Physical Layer. For instance, when the available memory is not sufficient for holding a “flattened” data block (line 2), the system computes the current hot/cold ratio (line 4) and if the current ratio is higher than the predefined ratio, data is evicted from the hot queue and moved to the cold queue. The system compresses the evicted data block from the hot queue and puts it into the cold queue (lines 6–10). Otherwise, if the current ratio is not higher than the predefined ratio, data is evicted from the cold queue. If the evicted data block is modified, the system writes it back to the Physical Layer.
**Algorithm 2:** Check available memory and Evict data from buffer if need
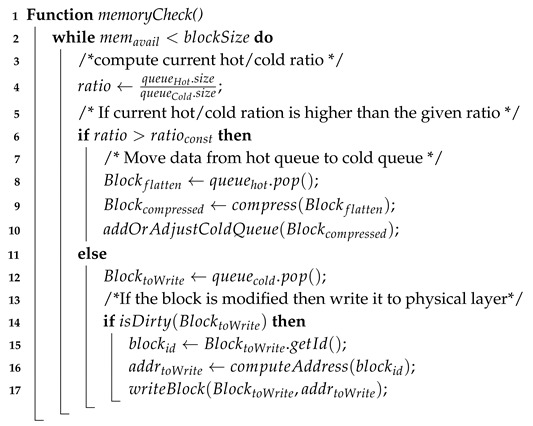


Dividing the buffer into a hot queue and a cold queue and to move data blocks between them is our first strategy to delay write operations when the system needs more memory. However, in the worst case, the Buffer Manager still has to write the data to Physical Storage. To make the Buffer Manager more efficient, we keep data blocks in each queue in the order illustrated in Figure 11. This ordering follows the following priorities:the clean/unmodified blocks;the higher density blocks, defined by the ratio between the number of triples in a block and the capacity of the block i.e., $density_{BlockA}=\frac{\#triples_{BlockA}}{capacity_{BlockA}}$;the least recently used blocks.

This prioritisation allows us to keep dirty/modified blocks in the buffer as long as possible to delay write operations and group more updates into dirty/modified blocks. In case we need to release memory, we always refer to the cold queue and start from the beginning of the priority list. Consequently, we prioritise clean/unmodified blocks to be evicted. Since we do not have to perform any write operation for them, we can just release the memory they occupy. Then, we prioritise blocks that contain many triples, i.e., high-density blocks, as they group multiple updates into one erase-write operation. Finally, the traditional LRU strategy is used to avoid writing the same block multiple times in a row.

## 7. Adaptive Strategy for Iterative Join Execution

The most resource-intensive task of answering a SPARQL query is to perform the graph pattern matching over the RDF dataset. The graph matching operator executes a series of join operations between RDF triples that match the triple patterns. Join operations have the greatest impact on the overall performance of a SPARQL query engine, typically requiring a large number of comparison operations that can only be done efficiently if records are stored in memory.

The join performance can be tuned by optimisation algorithms which plan optimal join orders and join algorithms [27,75,76]. These approaches assume that memory is always available during the course of the execution of a chosen query plan. However, in light-weight computing devices memory is critically low and, as such, the memory resources available to an RDF engine are unreliable, e.g., a surge of the number network connections to the device might drain available memory for all other running processes. Lack of memory may block join operations that require temporary virtual memory such as hash-joins or sorted-merge joins, and thus hurts the overall performance of the query engine or probably crashes the engine.

Materialisation techniques that write intermediate join results to storage are an attractive solution for the issue of memory shortage [68]. However, on flash storage, writing is much slower if a random write operation happens. Furthermore, only a limited number of erase operations can be applied to a block of flash memory before it becomes unreliable and fails.

To minimise the memory required for executing a SPARQL query and making the best use of the indexing scheme introduced in Section 5, we adopt the *one-tuple-at-a-time* approach to compute the join. This approach can reduce the memory consumption as no virtual temporary memory is required to buffer the intermediate join results. The basic idea of the algorithm to compute the join of a graph pattern is as follows: A mapping solution (mapping for short) is continuously sent to visit each triple pattern of the graph pattern. In each visit, it searches for triples matching the triple pattern. For each matched triple, variables in the triple pattern and the corresponding value in the triple are added to the mapping. The mapping with new values will be sent to visit the next triple pattern, or be returned as query result when all triple patterns have been visited. The pseudo code of the join propagation algorithm is given in Algorithm 3.
**Algorithm 3:** Join propagation
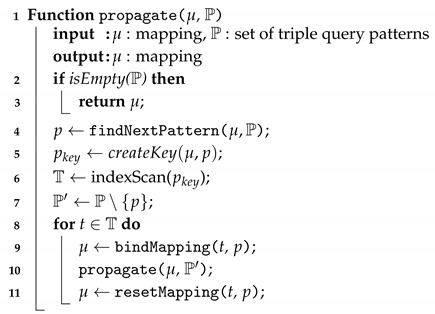


The propagate(μ, P) function is used to recursively propagate the input mapping. The function starts with an empty mapping μ and a set of unvisited triple patterns P. For each run, it checks and returns the input mapping as a result, if there is no triple pattern left to visit (line 1–2). Based on the given input mapping, it looks for the optimal unvisited triple query pattern to visit (line 3). To search for triples matching pattern *p*, an index search key $p_{key}$ is created by replacing the variables in *p* according to μ (line 4). For each matched triple *t*, the corresponding variables and values are bound into the mapping. Then another propagation of the mapping to the remaining unvisited triple patterns is initiated (line 7–10).

In each run of the propagation algorithm, the function findNextPattern(μ, P) is called to find the optimal triple pattern to execute the propagation (see Algorithm 4). For each triple pattern *p* in P, the set of triple query patterns, the function searches for a triple pattern that shares variables with the input mapping μ (line 4). With each shared pattern found, an index search key pattern, $p_{key}$, is created (line 5). An index lookup on $p_{key}$ is executed to search for the upper bound and lower bound positions of the set of the matching triples in the index, as described in the previous section (line 6). The size of the index lookup I is defined as the range between the upper bound and lower bound positions (lined 7–9). The function returns the triple pattern that has the minimal size of the index lookup (line 11).
**Algorithm 4:** Find next pattern
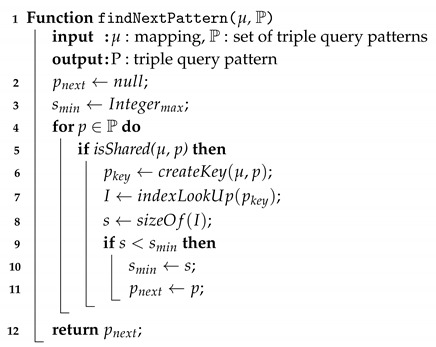


The join propagation algorithm is similar to the nested iterations. Nested loop join is often argued to have a poor performance as it does not attempt to prune the number of comparisons. However, supported by an efficient index scheme, an index nested loops join can perform as well as other join algorithms [73]. With the design of our storage, the index lookup can be done mostly within the buffer layer, only two extra I/O operations may be required. The visitor pattern, that sends a mapping to visit each triple pattern and to execute index lookup, reduces the extra memory for the joins as only a mapping is kept in the main memory. This mechanism also enables the adaptivity of the joins. The function findNextPattern(μ,P) decides which triple pattern the mapping should visit first. Similarly to a routing policy of stream processing engines, e.g., Eddies [74] or CQELS [77], this function defines the propagating policy to achieve a certain optimisation purpose. In our case, we attempt to minimise the number of propagations by choosing the shortest index scan in each run. Note that, this is also the key place for adding sophisticated optimisation algorithms, e.g., adaptive caching algorithm to be discussed in our future work.

## 8. Evaluation Results

Operating systems for a specific type of devices are usually customised and optimised to meet a specific hardware configuration. For example, by default, Raspbian is installed on the Raspberry Pi Zero while a Galileo Gen II is running the Yocto 1.4 Poky Linux distribution. A Java virtual machine is available on most edge devices and Java is platform independent. Hence, we choose to implement our approach in Java to take advantage of its “compile once run anywhere” property that enables the portability of our RDF4Led engine.

RDF4Led is developed by reusing Jena TDB code base and following the RISC-style design as presented in Section 4.1. We only selected the required components and re-implemented those with our algorithms (see Section 5, Section 6 and Section 7). As a result, the size of RDF4Led is only 4 MB, while the sizes of Jena TDB is 13 MB; the size of RDF4J is 58 MB; and the size of Virtuoso is 180 MB.

Figure 12 reports the result of **Exp1** in which we measure and compare the update throughput of RDF4Led against Jena TDB, RDF4J, and Virtuoso. The description of **Exp1** can be found in Section 3.4. Due to the similar result, we omit the experiment results for Raspberry Pi 2 (RPi3), and Beagle Bone Black (BBB), and report the experiment results on Intel Galileo Gen II (GII), Raspberry Pi Zero W (RPi0), and Raspberry Pi 3 (RPi3).

The results show that in comparison to Jena TDB and Virtuoso, RDF4Led can store a larger dataset and has a much higher update throughput. On GII, RDF4Led scaled up to 22 million triples, which is three times larger than Virtuoso and four times larger than Jena TDB. We stopped the experiment of RDF4Led on GII when its speed dropped below 80 triples/second after inserting 22 million triples. The scalability on Pi0 (see Figure 12b) and BBB was similar. On both these devices, RDF4Led was able to add the full 50 million triples. The update throughput of RDF4Led also decreased when the number of triples in the store increased. Among the three engines, RDF4Led has the highest inserting throughput on GII and RPi0. On GII, with 5 million triples inserted, the update throughput of RDF4Led was still around 350 triples/s, whereas Virtuoso was only able to insert data with a speed of 125 triples/s. On Pi0, RDF4Led performed update operations two to three times faster than Virtuoso. Even when the size grew to nearly 50 million triples, RDF4Led’s speed still stayed at 300–350 triples/s which was two times faster than the speed of Virtuoso with only 15 million triples. On RPi3, RDF4Led updated faster than Virtuoso in the beginning. When its storage size reached 20 million RDF triples, the speed of RDF4Led became lower than that of Virtuoso. However, RDF4Led still ran faster than RDF4J and Jena TDB.

The Exp1 results show how the lack of memory negatively influences the scale of standard RDF engines like Jena TDB, RDF4J, and Virtuoso. RDF4Led can insert more data as it has a smaller memory footprint and requires less memory to maintain the indexes. Furthermore, compared to the other engines, RDF4Led inserts data faster as our flash-aware index structure and writing strategy are better compatible with the flash memory’s I/O behaviours. Other RDF engines employ B${}^{+}$ tree to index RDF data in their storage. Their low throughput confirmed the negative influence of flash I/O behaviours on the write performance of such disk-based indexing techniques. Because RDF4Led was designed to save memory, on the devices with more RAM like RPi2 and RPi3, Virtuoso can run faster than RDF4Led. Virtuoso’s algorithms need bigger memory buffers to achieve their best performance.

The results of **Exp2** are shown in Figure 13. In this experiment, we compared the query response time of RDF4Led with that of three engines on GII, RPi0 and RPi3 with a dataset size that all engines could handle. The results show that RDF4Led answered all the queries considerably faster than Jena TDB and RDF4J on all devices. RDF4Led, RDF4J and JenaTDB follow the nested execution model to compute multiple joins between RDF triples that match triple patterns. However, Jena TDB and RDF4J were implemented with an iterator pattern, while RDF4Led followed the visitor pattern. In general, both algorithms execute lookup operations and index scan operations to extract the compatible triples from the dataset. The performance of these algorithms is mainly influenced by the performance of the lookup and index scan operations on the indexes. The better performance of RDF4Led indicates that our lightweight index structure helps RDF4Led outperform the B+ tree implemented in Jena TDB.

On the same dataset and on the same device, RDF4Led only answers the queries generated from templates F2 and S1 faster than Virtuoso does. These queries include star-shaped pattern of more than 6 triple patterns. In other cases, RDF4Led is slower than Virtuoso as it does not aggressively pre-allocate a fix amount of memory for sophisticated optimisation algorithms (Virtuoso allocates two to three times more than RDF4Led). We see this as an option to improve query performance in our future work. However, at the current stage, RDF4Led can deliver good performance for datasets of up to 50 million triples on these devices, e.g., 5 s at maximum and 1 s on average query execution times. The performance and scalability of RDF4Led can enable these kinds of low-capability devices to handle approximately 600 thousand sensor observations or 6 month worth of data of 25 weather stations in a single active RDF graph.

For **Exp3** the memory consumption of RDF4Led and the other engines is reported in Figure 14. In the insertion experiment, RDF4Led consumed only 85 MB of memory even when the storage went up to 50 million triples. In the query evaluation experiment, RDF4Led used less memory than the other engines did in the insertion test. Even with a dataset of 50 million triples, RDF4Led used only 80 MB. This was only a half of the memory that Jena TDB used in the query test with 10 million triples and was only 10% of the memory that the Virtuoso occupied constantly.

## 9. Conclusions

This paper presented an empirical study of the scalability of RDF engines running on resource-constrained devices, which are representative of typical IoT edge nodes. The results of the study show that these engines do not scale on such devices due to the lack of main memory and the special I/O requirements of flash storage. To address these problems, we proposed RDF4Led, a RISC-style approach to building an RDF engine tailored to resource-constrained edge devices. Our approach includes a flash-memory-aware storage structure, a flash-memory-aware buffer management strategy for RDF data and a low-memory-footprint join strategy for improved scalability as well as robustness. The RDF4Led engine is significantly smaller in terms of memory footprint than generic RDF engines like Jena TDB, RDF4J or Virtuoso. We tested RDF4Led on five different types of ARM boards. These experiments showed that RDF4Led can handle 2–5 times more data than its competition. Moreover, RDF4Led requires only 10%–30% of the memory consumed by Jena TDB, RDF4J and Virtuoso when operating on the same size of dataset. It can handle up to 50 million triples with approximately 115 MB of memory and can outperform its competitors in updating throughput; it is faster in answering queries than its Java counterpart, Jena TDB. While Virtuoso can deliver faster query processing time, it does so by pre-allocating a fixed amount of memory, which is 3 times more than that required by RDF4Led and with a significantly more complex implementation.

## Figures and Tables

**Figure 1 sensors-20-02788-f001:**
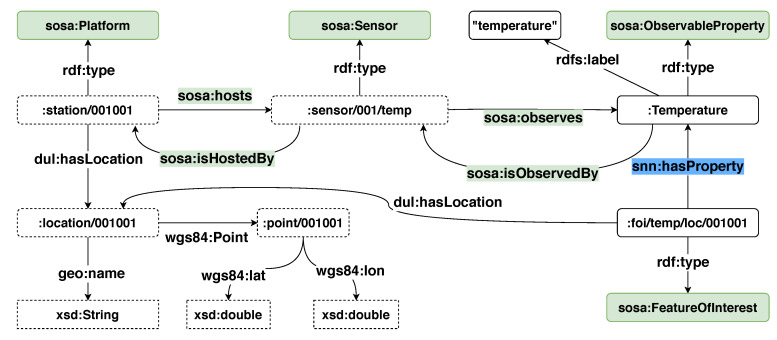
The RDF schema describing the weather stations in the ISD dataset.

**Figure 2 sensors-20-02788-f002:**
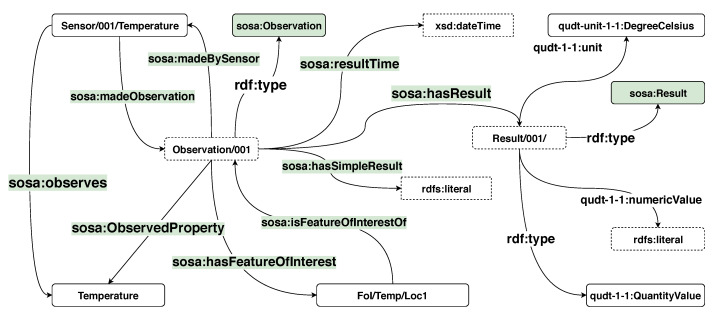
The RDF schema of observations.

**Figure 3 sensors-20-02788-f003:**
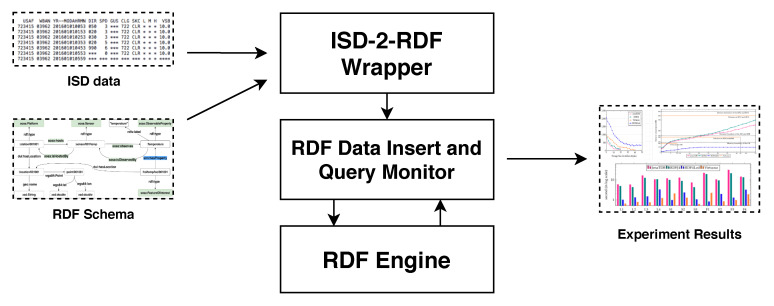
Setup of the RDF data insertion and querying experiments.

**Figure 4 sensors-20-02788-f004:**
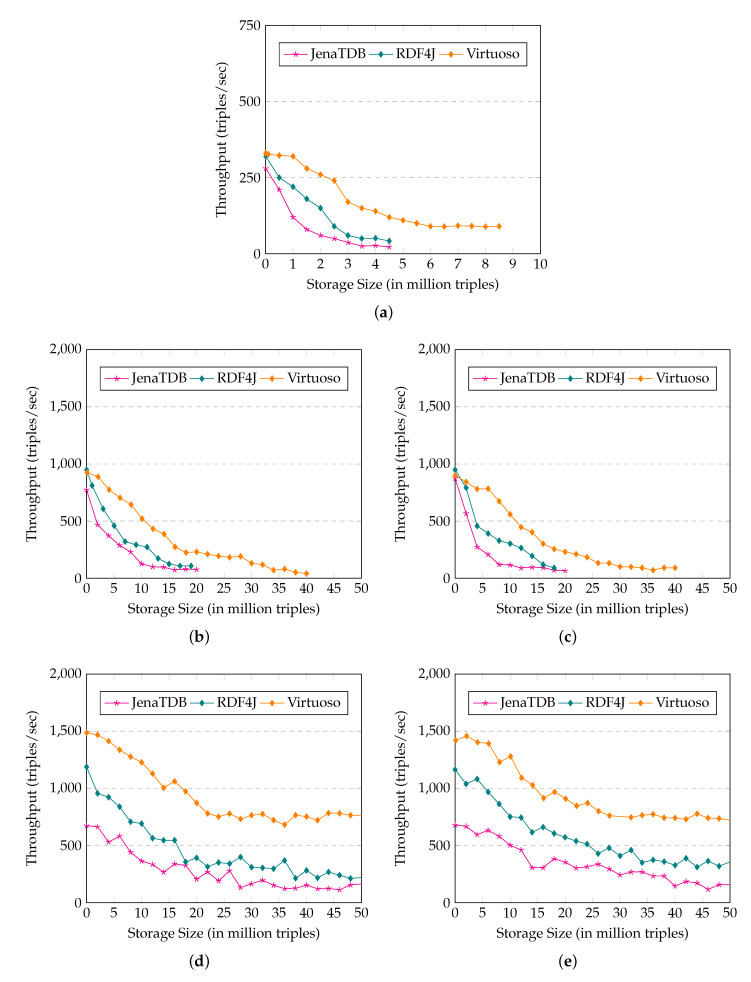
Inserting throughput test results of Jena TDB, RDF4J, Virtuoso on Gallileo Gen II, BeagleBone Black, Raspberry Pi Zero Raspberry Pi 2, and Raspberry Pi 3. (**a**) Insert throughput results on Gallileo Gen II; (**b**) Insert throughput results on Raspberry Pi Zero; (**c**) Insert throughput results on BeagleBone Black; (**d**) Insert throughput results on Raspberry Pi 2; (**e**) Insert throughput results on Raspberry Pi 3.

**Figure 5 sensors-20-02788-f005:**
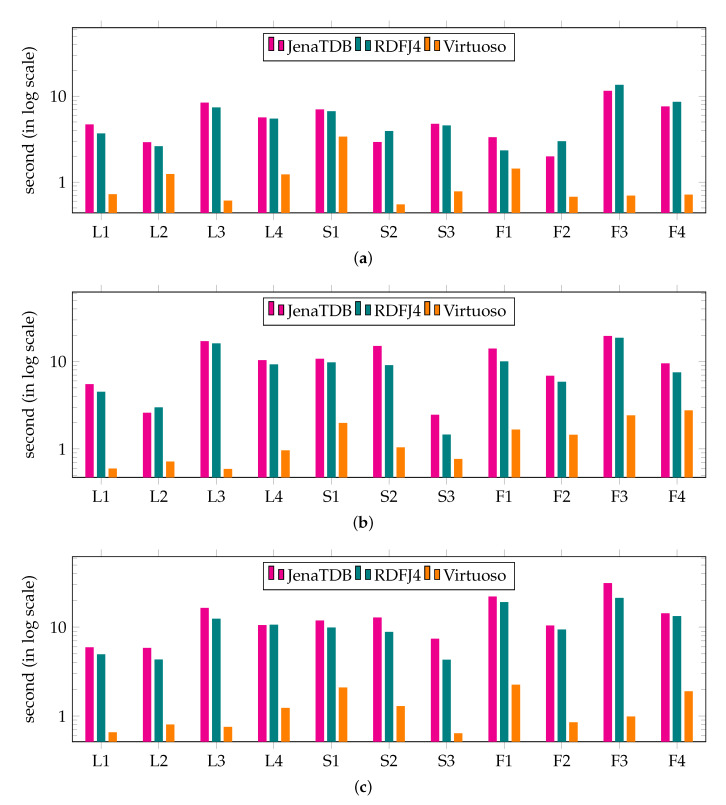
Querying test results of Jena TDB, RDF4J, Virtuoso and RDF4Led. (**a**) Query response time against 5 million triple dataset on Gallileo Gen II; (**b**) Query response time against 20 million triple dataset on BeagleBone Black; (**c**) Query response time against 50 million triple dataset on Raspberry Pi3.

**Figure 6 sensors-20-02788-f006:**
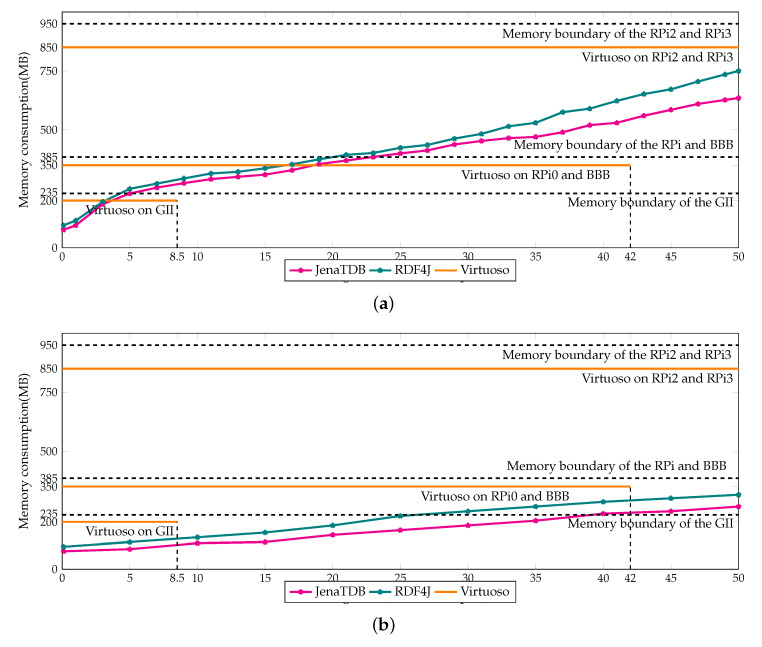
Memory consumption of RDF4J, Jena TDB, Virtuoso. (**a**) Memory consumption of RDF engines in update throughput test; (**b**) Memory consumption of RDF engines in query evaluation.

**Figure 7 sensors-20-02788-f007:**
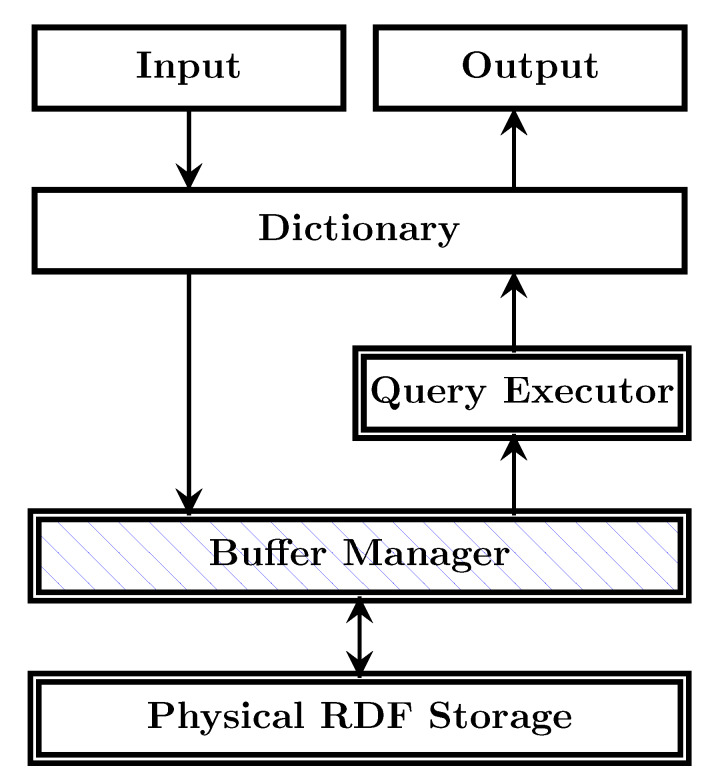
Architecture Overview.

**Figure 8 sensors-20-02788-f008:**
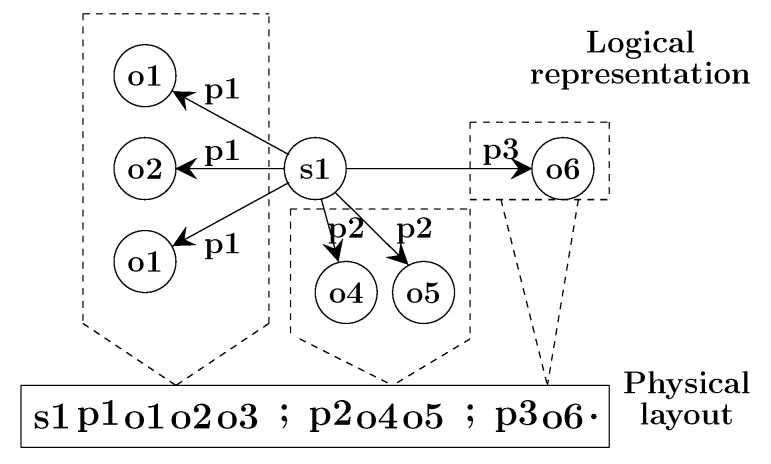
Example of an SPO molecule.

**Figure 9 sensors-20-02788-f009:**
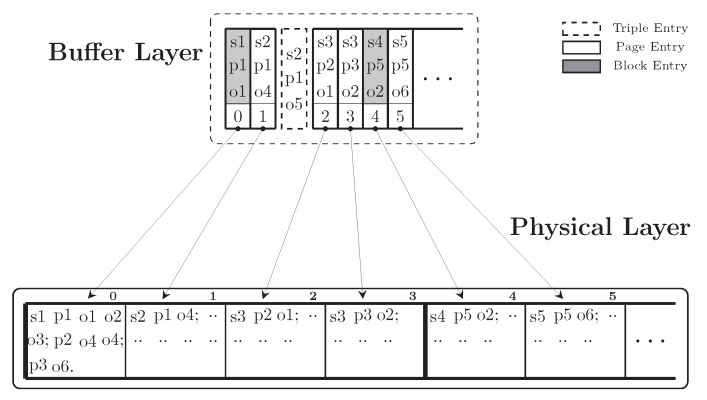
Two-layers storage model consist of a Physical Layer and a Buffer Layer.

**Figure 10 sensors-20-02788-f010:**
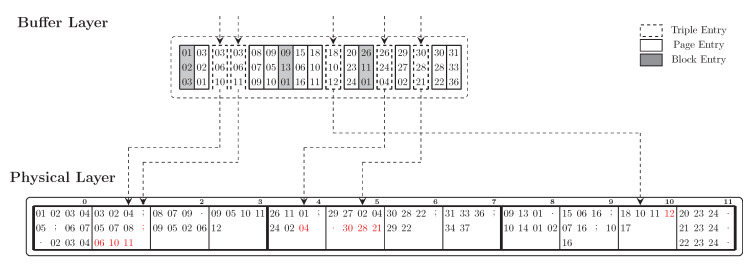
Example of clustering of writes.

**Figure 11 sensors-20-02788-f011:**
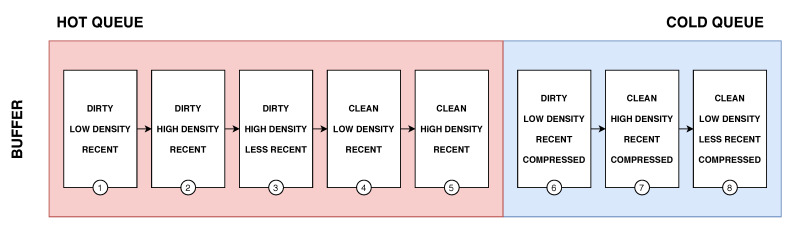
Example of how data block is organised in the buffer queues.

**Figure 12 sensors-20-02788-f012:**
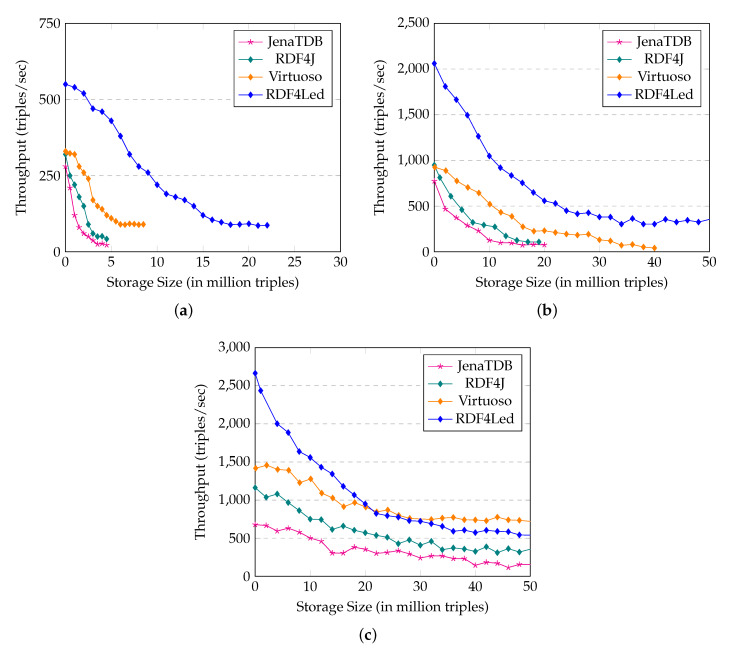
Inserting throughput of RDF4Led compared to Jena TDB, RDF4J and Virtuoso on Gallileo Gen II, Raspberry Pi Zero and Raspberry Pi 3. (**a**) Insert throughput results on Gallileo Gen II; (**b**) Insert throughput results on Raspberry Pi Zero; (**c**) Insert throughput results on Raspberry Pi 3.

**Figure 13 sensors-20-02788-f013:**
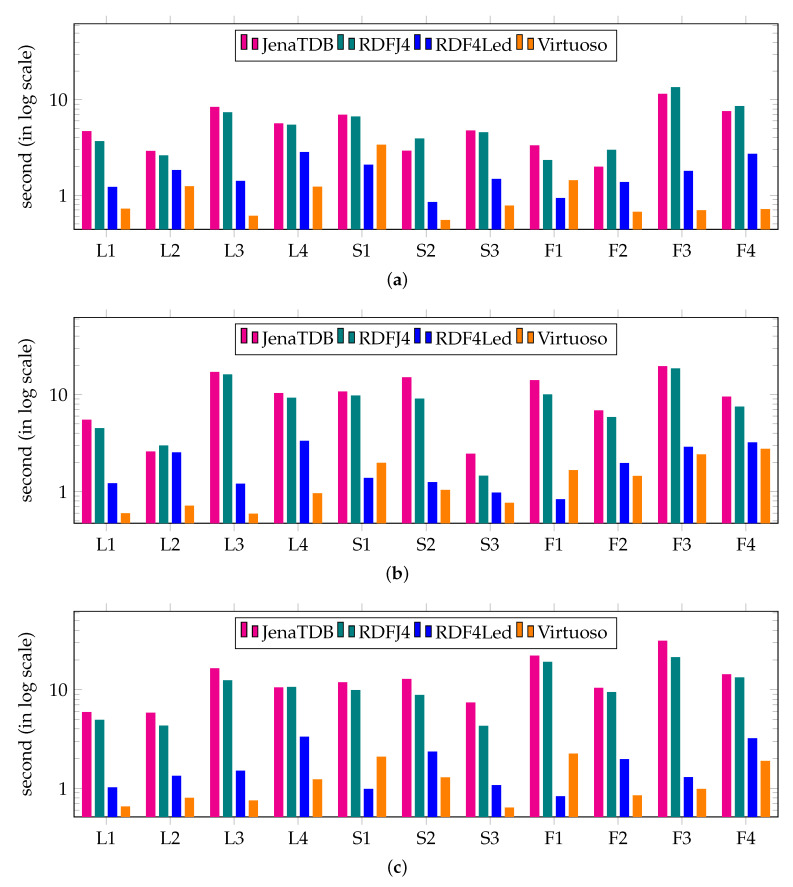
Query test results of Jena TDB, RDF4J, Virtuoso, and RDF4Led. (**a**) Query response time against 5 million triple dataset on Gallileo Gen II; (**b**) Query response time against 20 million triple dataset on Raspberry Pi Zero; (**c**) Query response time against 50 million triple dataset on Raspberry Pi 3.

**Figure 14 sensors-20-02788-f014:**
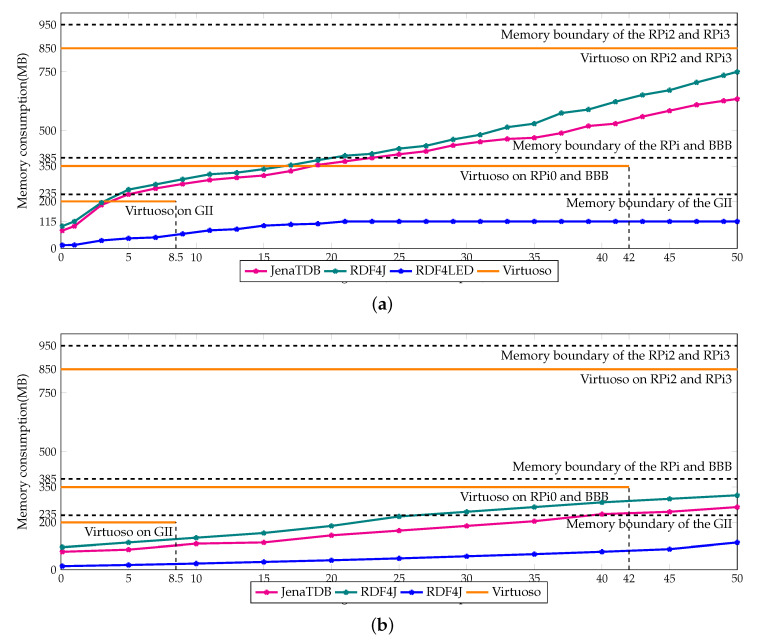
Memory consumption of Jena TDB, RDF4J, Virtuoso, RDF4Led. (**a**) Memory consumption report of update throughput test; (**b**) Memory consumption report of query evaluation test.

**Table 1 sensors-20-02788-t001:** Hardware configurations of the devices used in the experiments.

Device		GII	RPi0	BBB	RPi2	RPi3
Cost		65 EUR	15 EUR	55 EUR	35 EUR	45 EUR
CPU	model	Quark	ARM A8	ARM 11	ARM A7	ARM A53
	freq.	0.4 GHz	1.0 GHz	1.0 GHz	900 MHz	1.2 GHz
	n${}_{cores}$	1	1	1	4	4
RAM		256 MB	512 MB	512 MB	1 GB	1 GB
Storage		Transcend MicroSD 16GB class 10 (40 MB/s)				
OS		Yocto 1.4 Poky Linux Distribution	Rasp. Lite	Debian 7.0	Rasp. Lite	Rasp. Lite

**Table 2 sensors-20-02788-t002:** Characteristics of the RDF engines used in the experiments.

	Technical Characteristics				
	Developed in Language	Backend DB	File Access	Data Structure	Version
Jena TDB	Java	native store	File Caching	B${}^{+}$tree LRU Cache	3.14.0
RDF4J Native Store	Java	native store	n/a	BTree	3.1.0
Virtuoso Open-Source	C++	row store	n/a	B${}^{+}$Tree	6.1.8

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
