# Peer review of "Pushing the Scalability of RDF Engines on IoT Edge Devices†"

_sensors, 2020, doi:10.3390/s20102788_

Round 1

Reviewer 1 Report

In my opinion the work presented deserves to be published and it reports a good research work and experiment.

Nevertheless, the main issue is that it seems the authors didn't do a complete proof reading of the paper before submitting it and the paper contains a big number of typos which I think deserve a major revision. 

My advise is that the authors ask some help from an English expert.

Apart from that, the structure of the paper could probably be improved. The authors present some experiments with traditional Virtuosso, Jena and RDF4J in section 3.5, and later, in section 8 they present the evaluation results of the same systems against the RDF4Led proposed by the authors. I think it would suffice to present just the experiment and the comparison between the 4 systems as several figures contain repeated information which could be avoided: figures 3, 4, 5, 6, 7 and 8 are repeated adding only the values of RDF4Led in figures 14, 15 and 16.

I wonder if the implementation and the datasets employed for the experiments are available. It would be great to have a repository with the source code and those datasets to be able to reproduce the evaluation results.

I noticed that the authors have recently published a paper related to Stream processing https://link.springer.com/chapter/10.1007%2F978-3-030-41407-8_20 which was probably published after this paper's presentation to the conference but which could probably be cited as part of the future work.

In fact, the approach presented by the authors could be combined with a Stream based RDF validation as proposed in https://link.springer.com/chapter/10.1007/978-3-030-06149-4_6. Maybe the authors could consider this for future research.

Here is a list of some comments or typos that I found:

  • Line 29. a half extrabytes? or exabytes?
  • Line 31. ...of the huge amount information <of/on/ the IoT...
  • Line 37 ...an interesting study in IoT
  • Line 38. ...the better interpretation of data can be achieved by enabling machines intelligently...
  • Line 72 ...such <a/an> integration...
  • Line 88 ...<consumer/consume> less memory...
  • Line 121, it is not clear what is an entity...does it contain a blank node?
  • Line 125...made <an/a> wind...
  • Line 127...in other standard format
  • Line 128
  • Line 130 denoting <an/a> named entity...
  • Line 131...as a consequence there are 2 types of RDF triples...seems to ignore blank nodes again...are blank nodes supported?
  • Line 141...to query RDF data set...
  • Line 144 ...integration heterogeneous ...
  • Line 148 ...is RDF graph...
  • Line 157...variation of RDF
  • Line 159...<shap/shape>
  • Line 169...<fasten/faster>
  • Line 170...the evaluation of SPARQL...
  • Line 172...depends
  • Line 176...there are amount of work on...sounds strange
  • Line 200...RDF data is as
  • Line 202...can be
  • Line 211...are equipped large...
  • Line 218...are
  • Line 223...<The/Two> recent notable works...
  • Line 256...on these ...
  • Line 282...and actuators <can/to> be used ...
  • Line 282...therefore it can be used...
  • Line 296...was another well known...
  • Line 327...Virtuoso uses relational database...
  • Line 332...RDF datasets <is/are> stored as...
  • Line 346...<an/a> unique
  • Line 348 I think the use of < > for prefixed URIs is not necessary and would be better to just use station:001001...the same for the rest of URIs in the example description.
  • Line 369...the queries for the study <was/were> designed...
  • Line 373...the <queries/query> templates
  • Line 377...with three experiment
  • Line 378...on each <devices/device>
  • Line 418...on <these/the> other hand...
  • Line 435...50 millions RDF triples...
  • Line 467...are equipped lightweight...
  • Line 559...a hybrid data structure
  • Line 561...a set of <relates/related> resources...
  • Line 570. The sentence "In the buffer layer, we keep the information about the first triple of a molecule in a page and the first page in a block" was not clear to me. Maybe an example would help?
  • Line 577 it talks about "tuple entries" and the example of figure 11 only contains block and page entries, would it be possible to have an example with tuple entries?
  • Line 579...all triples are sorted...according to what?
  • Figure 11 would be more clear if it included a legend
  • Line 611...Note that these triples <contains/contain> only...
  • Line 619...at the same time to <same/save> expensive...
  • Line 747...the dataset <using/used>...
  • Line 758...on both devices
  • Line 805...<significant/significantly> smaller
  • Line 806...as much its counterparts
  • Line 825...<Start/Star> join queries

Author Response

Thank you for your valuable feedbacks. Here are our responses.

Point 1: I wonder if the implementation and the datasets employed for the experiments are available. It would be great to have a repository with the source code and those datasets to be able to reproduce the evaluation results.

Response 1:  Yes, We have updated a link to our online source code and datasets. (line 287 and line 408)

Point 2: I noticed that the authors have recently published a paper related to Stream processing https://link.springer.com/chapter/10.1007%2F978-3-030-41407-8_20 which was probably published after this paper's presentation to the conference but which could probably be cited as part of the future work.

Response 2: We have added some descriptions of our future works.

Point 3:  Line 121, it is not clear what is an entity...does it contain a blank node? 
Line 131...as a consequence there are 2 types of RDF triples...seems to ignore blank nodes again...are blank nodes supported?

Response 3: We mentioned “blank node” in line 145: Blank node is used for expressing an anonymous entity.

Point 4: Line 570. The sentence "In the buffer layer, we keep the information about the first triple of a molecule in a page and the first page in a block" was not clear to me. Maybe an example would help?

Response 4: We have added some more precise explanation in line 594 and line 595: “In the Buffer Layer, to keep the reference to a data page, we cache the first triple of the first molecule in the page. The reference to a data block is the first page of the data block.”

Point 5: Line 577: It talks about "tuple entries" and the example of figure 11 only contains block and page entries, would it be possible to have an example with tuple entries?

Response 5: We have updated the figure 11 as figure 10 in the revised version (page 18).

Point 6: Nevertheless, the main issue is that it seems the authors didn't do complete proofreading of the paper before submitting it and the paper contains a big number of typos which I think deserve a major revision. 

Response 6: We have corrected all the typos that the reviewer listed.

Reviewer 2 Report

This article describes how to enable scalable and robust RDF engines for IoT devices through an extensive evaluation. The evaluation includes sufficient data that might be useful for the future direction.

Some general comments:

  •   It would be easier to go through if you first say a few texts about RDF in the introduction.
  •   A list of your contributions in bullet points in the introduction may be useful for the readers.
  •  I would recommend adding a diagram for the experiment set up so that the readers could understand the architectural view to implementation and finally the results. All these should have good sync.
  •  I would avoid so many Listing X in the article as I have a reasonable doubt whether if they are interested to read these. In my opinion, this inclusion breaks the readability of the article. 
  • Line 27: The major challenges …. solved. I would recommend adding those challenges in a few sentences.
  • Finding links towards the contributions are difficult to follow.

Suggestions for Grammar:

Line 3: we have investigated?

Line 26: "," before for example.

Line 27: Please remove extra superscript after Gartner and elsewhere if any.

Line 157:  "and" before "o".

Minor typo:

Conclusion: Jena TDB vs Jena TBD?

Line 231: space before bracket?.. hardware configurations(eg. memory, CPU, storage)

Line 159: Complex shap? "," after e.g

Line 158: one space before "(" in many places.

I think that all of these must be corrected.

Author Response

Thank you for your valuable feedbacks. Here are our responses. 

Point 1: It would be easier to go through if you first say a few texts about RDF in the introduction.

Response 1: We have a background section briefly describing the RDF. We added some text to indicate the readers that RDF is presented in the background section.

Point 2:  A list of your contributions in bullet points in the introduction may be useful for the readers.

Response 2: Yes, we updated a list of our contributions on page 3.

Point 3: I would recommend adding a diagram for the experiment set up so that the readers could understand the architectural view to implementation and finally the results. All these should have good sync.

Response 3: Yes, we created a diagram for the experiment setup on page 11.  

Point 4: I would avoid so many Listing X in the article as I have a reasonable doubt whether they are interested to read these. In my opinion, this inclusion breaks the readability of the article. 

Response 4: Yes, we removed the appendix A. We added a link to our online repository where the readers can find the query used in our evaluation.

Point 5: Please remove extra superscript after Gartner and elsewhere if any.

Response 5: We have removed all the footnotes and moved the link to references.

Reviewer 3 Report

The paper presents an RDF engine designed and optimised for running on IoT typical hardware devices.

The paper is well written and organised.

The paper is an extension of a 2018 conference paper. several parts, including figures are in common between the papers. One possible improvement should be to change these parts an update some descriptions. for instance on row 300 where the raspberry Pi available models are mentioned, Pi 3 is missing, while it is used in the benchmark. Moreover even Pi 4 model B is not mentioned.

Considering the target audience of the journal the description and algorithms provided in sections 6 and 7 could benefit from pictures explaining some working details.  

In the valuation should be mentioned that the microprocessor of the raspberry PI 3 model B has a 64bit architecture. 

Author Response

Response to Reviewer 3 Comments:

Thank you for your valuable feedbacks. 

Point 1: For instance on row 300 where the raspberry Pi available models are mentioned, Pi 3 is missing, while it is used in the benchmark. In the valuation should be mentioned that the microprocessor of the raspberry PI 3 model B has a 64bit architecture. 

Response to 1: We added the description of PI 3 in line 320-322.

Round 2

Reviewer 2 Report

The sentences in the list of contributors are grammatically incorrect. These should be fixed. In addition, to my understanding, this list is incomplete based on the overall presentation of the work.

Table 1: I do not think that Yocto is an OS. It should be fixed.

What is the point to have line 837-844 in the conclusion? I would suggest adding these lines in the background not in the conclusion. The conclusion should highlight the summary of the article which follows the list of contributions included in the introduction.

Grammar:

General comment: The authors should carefully consider the grammars in the article. I would highly recommend doing proofread before the next submission.

E.g., Line  40, 268: "," needs to be used before "and". This rule should be maintained in the whole article.

Line 539: The line should be refactored, in particular the starting of the sentence.

Line 794: The results … is -> are

Line 837: I would suggest not to use the past tense.

… many more.

I would like to repeat my previous review comment:

" I would recommend adding a diagram for the experiment set up so that the readers could understand the architectural view to implementation and finally the results. All these should have good sync.

"

To add an additional comment, the diagram will illustrate where is the gateway, where is your IoT test network, etc.

Author Response

Point 1:
"I would recommend adding a diagram for the experiment set up so that the readers could understand the architectural view to implementation and finally the results. All these should have good sync."

To add an additional comment, the diagram will illustrate where is the gateway, where is your IoT test network, etc.

Response 1: We think it is not necessary to have a diagram to illustrate where the gateway is, because the experiments are conducted individually on each device. It is not necessary to create a network for such experiments. Furthermore, the readers may be concerned of the network communication which is not in the scope of the experiment. When we introduced the experiment design, we cited the paper [32] to refer the reader to such architecture.

32. Desai, P.; Sheth, A.; Anantharam, P.  Semantic gateway as a service architecture for iot interoperability. 2015 IEEE International Conference on Mobile Services. IEEE, 2015, pp. 313–319.

Point 2: The sentences in the list of contributors are grammatically incorrect. These should be fixed. In addition, to my understanding, this list is incomplete based on the overall presentation of the work.

What is the point to have line 837-844 in the conclusion? I would suggest adding these lines in the background not in the conclusion. The conclusion should highlight the summary of the article which follows the list of contributions included in the introduction.

Response 2: We have updated the introduction and the conclusion as the reviewer suggested.

Point 3: The authors should carefully consider the grammars in the article. I would highly recommend doing proofread before the next submission.

E.g., Line  40, 268: "," needs to be used before "and". This rule should be maintained in the whole article.
Line 539: The line should be refactored, in particular the starting of the sentence.
Line 794: The results … is -> are
Line 837: I would suggest not to use the past tense.

Response 3: we have fixed these mistakes and carefully proofread the paper. 

Round 3

Reviewer 2 Report

I am happy with the improvement. 

The authors have made significant improvements as per my understanding.